# Assessing Risks to Wildlife from Free-Roaming Hybrid Cats: The Proposed Introduction of Pet Savannah Cats to Australia as a Case Study

**DOI:** 10.3390/ani9100795

**Published:** 2019-10-14

**Authors:** Christopher R. Dickman, Sarah M. Legge, John C. Z. Woinarski

**Affiliations:** 1National Environmental Science Program Threatened Species Recovery Hub, Desert Ecology Research Group, School of Life and Environmental Sciences, University of Sydney, Sydney, NSW 2006, Australia; 2National Environmental Science Program Threatened Species Recovery Hub, Centre for Biodiversity and Conservation Science, University of Queensland, St Lucia, QLD 4072, Australia; sarahmarialegge@gmail.com; 3Fenner School of Environment and Society, The Australian National University, Canberra, ACT 2601, Australia; 4National Environmental Science Program Threatened Species Recovery Hub, Research Institute for the Environment and Livelihoods, Charles Darwin University, Casuarina, NT 0909, Australia; john.woinarski@cdu.edu.au

**Keywords:** decision–framework, *Felis catus*, hybrid, *Leptailurus serval*, precautionary principle, predation risk, savannah cat, threatened species

## Abstract

**Simple Summary:**

The domestic cat, *Felis catus*, is often cross-bred with other species in the cat family to produce hybrid or ‘designer’ cats that are sought by people as pets. However, hybrid cats are often surrendered to wildlife shelters, or released, which leads to concern that they may establish free-roaming populations and damage native wildlife. In 2008, the Australian government rejected an application, on precautionary grounds, to import savannah cats (hybrids of the domestic cat and serval *Leptailurus serval*) into the country. We review the limited information informing this decision and then present a framework that identifies the native mammal species likely to have been most at risk of predation from savannah cats if importation and establishment had occurred. Assuming that savannah cats hunt similar prey to those that are hunted by both parent species, we estimate that 91% of Australia’s extant terrestrial mammal fauna would likely face some risk of predation from savannah cats, including 93% of non-volant mammal species that have threatened conservation status. The framework results strongly validate the decision to ban savannah cats from Australia. We suggest that our framework approach could be adapted to assess the likely risks that are posed by the arrival of other hybrid cats or hybrids of other animals.

**Abstract:**

Hybrid cats—created by crossing different species within the family Felidae—are popular pets, but they could potentially threaten native species if they escape and establish free-roaming populations. To forestall this possibility, the Australian government imposed a specific ban on importation of the savannah cat, a hybrid created by crossing the domestic cat *Felis catus* and serval *Leptailurus serval*, in 2008. We develop a decision–framework that identifies those species of non-volant native mammals in Australia that would likely have been susceptible to predation by savannah cats if importation and establishment had occurred. We assumed that savannah cats would hunt ecologically similar prey to those that are depredated by both the domestic cat and the serval, and categorised native mammals as having different levels of susceptibility to predation by savannah cats based on their size, habitat range, and behaviour. Using this framework, we assessed savannah cats as likely to add at least 28 extant native mammal species to the 168 that are known already to be susceptible to predation by the domestic cat, posing a risk to 91% of Australia’s extant non-volant terrestrial mammal species (n = 216) and to 93% of threatened mammal species. The framework could be generalised to assess risks from any other hybrid taxa.

## 1. Introduction

Hybridisation is a common occurrence in the animal kingdom, with at least 10% of closely related species estimated to hybridise in some parts of their geographical ranges [1]. The phenomenon is predicted to more frequently occur in the future as climate change drives shifts in species’ distributions and causes formerly isolated species to come into contact [2]. Within the cat family, Felidae, hybrids naturally occur between some members of the genus *Felis* (e.g., *F. catus* and *F. lybica* [3]); they might also occur between members of other genera, such as *Panthera* (e.g., *P. leo* and *P. tigris*), but in these cases crosses usually occur under captive conditions when mate choice is constrained [4]. Hybridisation has commonly occurred throughout the evolutionary history of the Felidae, with the result that some interspecific and intergeneric relationships remain poorly resolved [5]. Hybridisation can usually be recognised when viable hybrid offspring are produced by interspecific parents, and introgression when new genetic material is integrated from one species into another via backcrossing [6]. Both processes operate within the Felidae, although hybridisation is most frequently reported [7,8].

In the wild, hybridisation among felids has been most extensively documented between the domestic cat *F. catus* and the European wildcat *F. silvestris*, with hybridisation being considered to be a key threat to the integrity of ‘pure’ wildcat populations in parts of this species’ range [9]. In Scotland, for example, hybridisation appears to have been ongoing between the two species for many years, and hybrids and their descendants are now difficult to distinguish from wildcats on both morphological and genetic criteria [10,11]. Elsewhere in Europe the degree of hybridisation between *F. catus* and *F. silvestris* differs regionally [12,13,14], but it can be difficult to quantify owing to high levels of genetic and morphological variation in both taxa between regions [15,16]. Nonetheless, there is evidence that hybridisation is limited if the two species vary in their use of habitats or space [17]; enhancing such segregation has been proposed as a measure to reduce hybridisation and conserve *F. silvestris* in parts of Scotland [11].

Under captive conditions, *Felis catus* has been deliberately cross-bred with other *Felis* species, and with members of other felid genera, to produce hybrids with characteristics that are deemed to be desirable for the domestic pet trade. Interspecific crosses include *F. catus* with the jungle cat *F. chaus* to produce the ‘chausie’, and with the Indian desert cat *F. lybica ornata* to produce the ‘Punjabi’. Intergeneric crosses include *F. catus* with the leopard cat *Prionailurus bengalensis* to produce the ‘Bengal’ and with the fishing cat *P. viverrinus* to produce the ‘jambi’ or ‘viverral’. Other hybrids include crosses of *F. catus* with the margay *Leopardus wiedii* to create the ‘Bristol’ and with Geoffroy’s cat *L. geoffroyi* to obtain the ‘safari’ [18,19]. Members of at least three further genera within the Felidae are also sometimes crossed with *F. catus* to create designer hybrids for the pet trade and, less usually, crosses that do not involve *Felis* spp. might also occur, such as that between the margay and the ocelot *L. pardalis* to produce the hybrid ‘marlot’ [19].

Domestic cats have been selectively bred over many generations to exhibit varied hair length and density, coat colouration and pattern, face shape, length of ears, limbs, and tail, as well as traits, such as personality [20]. Hybridisation provides a quick way to introduce new traits into domestic cats for people who seek novelty or different kinds of beauty or behaviour in their pets. For example, the large (5–7 kg) and popular Bengal cat exhibits a spectacular spotted or rosette coat, and it is reputed to be playful and mischievous [18]; the chausie has relatively long limbs and it is prized for its agility and intelligence. However, hybridisation can also bring a host of problems. Species such as the margay and fishing cat have declining numbers in the wild, and the depletion of their populations to support the hybrid cat trade might represent another threat to their long-term persistence. Domestic cats have 38 chromosomes, whereas most felid species with which domestic cats are hybridised have only 36 chromosomes, frequently leading to reduced fertility or problems with gestation and live birth [19]. Hybrids may also retain the wild-type behaviours of their non-domestic ancestors and engage in aggressive interactions with their owners and other pets, or unwanted territorial behaviours, such as vocalising and urine-spraying to mark territory boundaries [18]. In consequence, hybrid cats might be deliberately released (as well as accidentally) by their owners, and some wildlife shelters report that hybrid cats are being surrendered to them at ‘alarming’ rates (although numerical data are scant), and are overwhelming their capacity to house or re-home these erstwhile pets (e.g., https://www.wildcatsanctuary.org/education/species/hybrid-domestic/what-is-a-hybrid-domestic/, accessed on 1 June 2019).

The welfare of surrendered cats (and other companion animals) is an issue of some concern, and it can lead to stress and anxiety when these erstwhile pets are displaced from their familiar surroundings, dietary and activity routines, and enter new social milieu; euthanasia is the ultimate fate for many [21]. However, a further problem is less obvious: many unwanted cats are not surrendered to shelters and are set free in parks and other habitats where they may have damaging effects on native species [22,23]. Individual cats have been documented to have strongly deleterious impacts on populations of prey organisms [24,25,26], and at the continental scale in Australia and the United States of America hundreds of millions to billions of terrestrial vertebrates are collectively killed by cats each year [23,27,28]. It is not certain whether hybrid cats that have been dumped or that have established free-ranging populations would exert significant additional effects on native fauna. Nor is it clear whether legislation that has been enacted to manage the effects of *Felis catus* would extend to cover cat hybrids; the status of hybrids in conservation and management programs is often clouded and contentious [29,30].

Here, we describe a case study in which the Australian government pre-emptively moved to prevent the importation and sale of a hybrid cat taxon that would have been a new addition to the country’s pet industry: the ‘savannah cat’, a cross between *Felis catus* and the serval *Leptailurus serval*. We first document the circumstances and sources of information that led to the decision to impose the ban on this particular hybrid, and describe some of the consequences. Subsequently, we propose a decision–framework that could be used to evaluate the potential risk to native fauna that is posed by the importation of hybrid cats—and, by extension, other novel taxa—using the savannah cat as an exemplar.

## 2. The Savannah Cat

Hybrid cats can be sold and kept as pets in some parts of the world, but in Australia the importation of such animals is tightly regulated by the Environment Protection and Biodiversity Conservation Act 1999 (EPBC Act) and the Biosecurity Act 2015. Hybrids (and nonindigenous species generally) can only be imported into the country if they are on the Live Import List of the EPBC Act or if they satisfy other requirements, although application can be made to the Minister for the Environment to include new taxa. Currently, only the mule, which is a hybrid between the horse *Equus caballus* and donkey *E. asinus* is on the Live Import List. However, as an historical exception, Bengal cats are allowed to be imported if individual animals can be shown by official pedigree papers to be removed by five generations or more from their leopard cat ancestor (http://www.environment.gov.au/biodiversity/wildlife-trade/live/import-list/hybrid-animals-guidance). The Live Import List specifically excludes only one hybrid: the savannah cat.

Savannah cats bear a general resemblance to their serval forebears, and have a tawny coat with dark spots, long ears and limbs. They are appreciably larger than domestic cats, but vary considerably in size. F_1_ males can weigh 8–11 kg (exceptionally up to 18 kg) and stand 40–45 cm high at the shoulder, whereas F_4_ or F_5_ males weigh 6.3–8.2 kg and stand 32.5–38 cm at the shoulder; females are slightly smaller [31]. The animals are active, terrestrial hunters, and agile climbers, which are able to leap 2.5 m high from a standing position [31]. These traits, together with a reportedly sociable and playful temperament, combine to make savannah cats popular in the designer pet trade.

### 2.1. Importation vs Prohibition of the Savannah Cat

In 2006, a small company based at Benowa in south-eastern Queensland—Savannah Cats Australia—gained preliminary permission from the Australian government to import 14 savannah cats from breeders in the United States of America. The animals were to be maintained in escape-proof facilities and bred for domestic sale. However, at the 14th Australasian Vertebrate Pest Conference (AVPC) in Darwin, 10–13 June 2008, reports circulated that the Australian government was preparing a paper to assess the risks that are associated with the importation of savannah cats, providing the first public acknowledgement that imports of these hybrids were being considered [32]. Attendees at the AVPC, including scientists whose work had done much to uncover the damaging effects of domestic and free-ranging cats on Australia’s native fauna [33], were concerned that savannah cats could have additional deleterious impacts on that fauna and they made this clear in statements on radio and in print media [34]. The government’s draft report was published just after the conference, and proposed an amendment to the EPBC Act to prohibit the entry of savannah cats into the country [35]. Although allowing just 20 business days—until 17 July 2008—for public comment on the draft report, the government received 529 submissions supporting a ban on importation and 24 in favour of allowing entry [36]. Among the respondents supporting the ban were organisations such as the Invasive Species Council, Invasive Species Cooperative Research Centre and the Wildlife Preservation Society of Queensland, with each presumably representing the views of their respective memberships [32]. On 24 July 2008, the Australian government published its final assessment report; this incorporated points that were raised during the public consultation process and recommended against importation [36].

Despite Savannah Cats Australia threatening to seek up to $A2 million in compensation if the hybrid cats were banned, on 3 August 2008 the then Environment Minister, Mr Peter Garrett, proceeded with an amendment to the EPBC Act to specifically prohibit the importation of savannah cats. The importation proponents challenged this decision, but the Federal Court upheld the ban on 29 June 2012, with costs awarded in favour of the Australian government (Parker v Minister for Sustainability, Environment, Water, Population, and Communities (2012) FCAFC 94; [32]). The Australian government’s final assessment report concluded that savannah cats would pose a grave—albeit generalized—risk to native fauna if they were to establish in the wild [36]. As the proposal to import savannah cats identified great risk with respect to animals escaping, establishing, and harming native fauna, the ministerial decision to ban them appears to be defensible and well-founded.

### 2.2. The Information Base for the Decision

The decision to ban the entry of savannah cats into Australia was made quickly: little more than six weeks elapsed from the time when the importation proposal became public knowledge to when it was rejected. The decision was a landmark and it catalysed a protocol for banning the importation of all new hybrids. However, the decision was not based on direct evidence regarding the impacts of savannah cats (or other hybrid cats) on prey populations, as explicit information on such impacts was not available from Australia or elsewhere in 2008 [36,37], nor has it been adduced in recent works [23,38]. Instead, the probable risks of importation were considered to be too great to allow the entry of savannah cats. The precautionary approach towards reducing risk in the absence of full scientific certainty has been well articulated, especially in situations where the risks might result in serious or irreversible damage [39,40].

Following protocols that were developed by Bomford [41,42], three broad categories of risk were considered by the Australian Department of Environment in evaluating whether savannah cats should be allowed entry to Australia: their likelihood of escaping captivity and harming people, their likelihood of establishing feral populations, and their potential for impact on native fauna if establishment occurred. In its final assessment report [36], the Australian government summarised information regarding the proposed security of breeding and holding facilities for savannah cats and on the behaviour of owners of savannah cats and other hybrid cats in other parts of the world. The report concluded that there was an extreme risk of escape or release and that individual escapees could pose a moderate danger to people [36]. Climate models were used to predict the potential distribution of both the serval and the domestic cat, and hence to map the potential distribution of the hybrid savannah cat if it were to establish. These models suggested that suitable climates exist for the serval across most of central and northern Australia [31], and, using relaxed model assumptions, most of the southern continental regions too [36]. Domestic cats occur across a very wide range of terrestrial habitats. Assuming that savannah cats might ultimately occupy a range between those that are predicted for the two parent species, the establishment of this hybrid cat could result in it roaming over up to 97% of Australia. In consequence, the government’s final assessment report concluded that the risk of establishment of feral savannah cats was extreme [36].

What would be the potential impact of the savannah cat on Australia’s native fauna if it were to become broadly established? This question was answered very generally in the final assessment report [36], with the conclusion being that a wide range of native species could suffer negative impacts from savannah cat predation. In seeking a more precise answer to the question, it might be reasonable to assume that any impacts of the hybrid would range between those that the two parent species are documented to have (e.g., [43]). We used this assumption, in particular examining the range of prey that might be susceptible to savannah cat predation, as the basis for developing our decision–framework.

### 2.3. Diets and Potential Impacts of Domestic Cats, Servals and Savannah Cats

Free-ranging and pet domestic cats in Australia depredate at least 747 species of native terrestrial vertebrates (151 native mammal species, 338 native bird species, and 258 reptile species) and they are estimated to kill 2.2 billion individual vertebrates in these three classes each year [28,44,45]. Most prey are small (<220 g), but on occasion prey can be almost as large as the cats themselves (4–5 kg; [46]). Most habitats can be occupied, although the densities of free-roaming domestic cats are higher in arid than in temperate environments following heavy rains, and densely-vegetated swamp and marshy habitats tend to be avoided [47]. Domestic cats are also vectors for parasites such as *Toxoplasma gondii* and *Toxocara cati*, which cause disease in humans, livestock, and wildlife, and they potentially compete with native predators for food and other resources [23]. Despite the large death toll that is exerted by domestic cats on native fauna, few studies have experimentally quantified the impacts of these cats on extant populations of prey and other species with which they interact. Nonetheless, in Australia a broad range of historical, comparative, correlative, and observational evidence indicates that the extinction of about 20–25 native mammals was largely driven by free-roaming domestic cats, with many extant species remaining at risk [23,48].

Very little information is available on the impact of servals on fauna in their native range in Africa. However, their diet includes a wide range of small and medium-sized vertebrates (frogs, reptiles, birds, and mammals), with larger prey, such as cane-rats *Thryonomys swinderianus* (up to 8 kg), featuring prominently and small antelope (up to 18 kg) being included on occasion [49]. Servals are mostly active by night and are efficient but occasional climbers, ascending to 4 m with ease and up to 9 m if pursued by larger predators [49,50]. They prefer to hunt in densely vegetated and even swampy areas, sometimes among aquatic grasses in water that is 50–80 mm deep [49]. Areas of high rainfall are preferred, and close proximity to water appears to be essential [49,51].

It is reasonable to suggest that the potential impacts of hybrid savannah cats on native prey would be greater than those of either parent species on its own, as servals can subdue larger prey, are accomplished and agile climbers, and prefer wetter and more densely-vegetated habitats than domestic cats. It follows that many Australian wildlife species that are currently secure in densely-vegetated habitats and swamps (habitats little used by domestic cats) would be exposed to increased rates of predation.

## 3. A Decision–Framework to Evaluate Risks of Hybrid Cats to Native Fauna

Combining some of the key behavioural and morphological traits of parent species might provide general insight into the range of those traits that could manifest in first and later generation hybrids, but does little to identify which native faunal species would be most at risk if the hybrid taxon became established. Below, we outline a framework that permits the identification of at-risk species, thus providing a more quantitative evaluation of the overall risk that might be posed by the arrival of a novel hybrid in any system, and insight into the taxa that would need to be targeted for conservation if establishment occurred. Although the framework should have general utility, we describe the potential impact of the savannah cat in Australia as a case study.

### 3.1. Developing the Framework

Early studies in Australia showed that the susceptibility of native species to predation by domestic cats could be predicted on the basis of several attributes of their biology. These attributes included the species’ body size, preferred habitats, behaviour, mobility, and fecundity, as well as the estimated density of cats within their geographical ranges [37,52]. An overall assessment of risk could be defined by assigning a rank score describing the intensity of cat-impact associated with each attribute (0 = no impact, 3 = high impact). Thus, a small mammal (<220 g) with nocturnal, terrestrial habits that occupied open habitat would be rated as being at high risk of predation from domestic cats and receive a high overall score, whereas a large, arboreal, or day-active mammal would be at little or no risk and receive a correspondingly low score. More-recent studies have incorporated further information regarding the population-level responses of native prey species to introduced predators from a wide range of published and unpublished sources, including expert opinion, to place species into one of four categories of predator-susceptibility: extreme, high, low, and not predator-susceptible [53]. *Extremely susceptible* species cannot persist in the presence of introduced predators unless, exceptionally, the predator is at very low density; *highly susceptible* species may persist in the short-term (e.g., 20 years) in the presence of introduced predators, but with severely reduced population size or viability; species with *low susceptibility* to introduced predators are likely to persist with these predators but with somewhat reduced population sizes or viability than where the predators are absent; and, *not-susceptible* species are unaffected by predator presence [53].

In the analyses of Radford et al. [53], all 246 non-volant species of Australian terrestrial mammals were assessed for their population-level susceptibility to predation by the domestic cat (Table 1), as well as to predation by the introduced red fox *Vulpes vulpes*. Ground-dwelling species were found to be more predator-susceptible than arboreal species and small to medium-sized species (adult female body weight 35 g–3.5 kg) more susceptible than their smaller or larger counterparts (also characterised by more detailed modelling of Australian native mammals in cat dietary collations: [54]). Notably, predator-susceptibility was strongly associated with the conservation status of species as assessed by the IUCN (including taxonomic updates documented in Woinarski et al. [48]): predator-susceptible species were much more likely to be threatened or now extinct, whereas species that were rated as not-susceptible or of low susceptibility to predation by introduced predators were more likely to be of least concern.

Here, we extend the framework and results of Radford et al. [53] to include native mammal species that would likely be susceptible to predation by the savannah cat were it to become established. Based on documented aspects of the diet and ecology of the serval (e.g., [49,50,55,56]) and of pet and free-roaming domestic cats (e.g., [23,38,57]) and the likelihood that their hybrids (savannah cats) would hunt prey with similar characteristics to—or intermediate between—both parent species [43], we make two assumptions. First, species that are assessed as susceptible to predation by domestic cats already (Table 1) would also be susceptible to predation by savannah cats; and second, additional species would be susceptible to predation by savannah cats if they weigh between 3.5 and 18 kg (female body weight), and/or are at least partly arboreal, and/or use densely-vegetated swamp, riparian, or wet forest habitats. For the second assumption, we consider that species would have high susceptibility to predation by savannah cats if they weigh 3.5–8 kg (female body weight) or are active up to 4 m above ground in trees, and to have low susceptibility if they weigh >8–18 kg (female body weight) or are active >4–9 m above ground [49,50]. In addition, species confined to wet or densely vegetated habitats, or with large parts of their geographical ranges in these habitats, were considered to be highly susceptible to predation by savannah cats if the body weight of females lies between 35 g and 8 kg. All the species were reviewed *post hoc*, and assessments adjusted to account for anti-predator defenses, such as spines or pugnacious behaviour (Appendix A). We did not use the ‘extremely susceptible’ category of Radford et al. [53] in addressing our second assumption, owing to uncertainty regarding whether any native species would be locally or regionally extirpated in the presence of savannah cats. Hence, our tally of native species that are susceptible to predation by savannah cats should be seen as conservative.

### 3.2. Using the Framework

Of the 246 non-volant mammal species that were assessed in the original analysis of Radford et al. [53], 26 were categorised as extremely susceptible to domestic cat predation, 41 as highly susceptible, 127 as of low susceptibility, and 49 as not susceptible; three now-extinct island endemic species were not assessed (Table 1). Overall, the original analysis found 168 extant species to be susceptible to predation by domestic cats (Table 1). The potential additional risks of predation by the savannah cat were assessed as affecting 61 extant species, moving them either from the category of not susceptible to domestic cat predation to being at low (n = 22 species) or at high risk (n = 6 species) from savannah cat predation, or from the category of low susceptibility to domestic cat predation to high susceptibility to savannah cat predation (n = 33) (Appendix A). These changes led to an almost two-fold increase in the number of species categorised as highly susceptible to predation, from 41 for domestic cats to 80 for savannah cats, and to correspondingly fewer species categorised as of low susceptibility and not susceptible to savannah cat predation (Table 1). After combining the IUCN categories Critically Endangered, Endangered, and Vulnerable into ‘threatened’, conservation status was strongly associated with species’ level of susceptibility to predation by savannah cats (χ^2^ = 134.4, degrees of freedom = 9, *p* < 0.001). Overall, 196 of Australia’s 216 extant non-volant terrestrial mammal species (91%) were assessed as likely to be at low, high, or extreme risk of predation by savannah cats, as were 54 of 58 species (93%) with IUCN status of Critically Endangered, Endangered, and Vulnerable (Table 1).

Species and species-groups that were rated as being more susceptible to savannah cats than to domestic cats included the semi-aquatic platypus *Ornithorhynchus anatinus* and water rat *Hydromys chrysogaster*, a broad range of rainforest specialists, such as the musky rat-kangaroo *Hypsiprymnodon moschatus*, *Uromys* spp., some *Melomys* spp. and semi-arboreal tree-kangaroos *Dendrolagus bennettianus* and *D. lumholtzi*, and mid-sized and larger marsupials, such as the spotted-tailed quoll *Dasyurus maculatus*, Tasmanian devil *Sarcophilus harrisii*, short-nosed bandicoots *Isoodon* spp., and wallabies, such as the parma wallaby *Notamacropus parma*, tammar wallaby *N. eugenii,* and pademelons *Thylogale* spp. (Appendix A). Species that were unlikely to be susceptible to predation by savannah cats were canopy-dwelling gliders and possums, such as *Petauroides* spp. and *Pseudochirulus* spp., large kangaroos and wallaroos *Macropus* and *Osphranter* spp., and the heavy-bodied (>18 kg) wombats *Vombatus ursinus* and *Lasiorhinus* spp. (Appendix A).

As noted above, our categorisations of species at risk from predation by savannah cats are conservative, and arguments could be made that further species of native mammals would be more susceptible to predation from savannah cats than from domestic cats. For example, we assessed four species of rock-wallaby *Petrogale* spp. that were not considered to be susceptible to predation from domestic cats as being at a low risk of predation by savannah cats (Appendix A). However, it is plausible that these species, and all other *Petrogale* spp. rated as having low susceptibility to predation by the domestic cat [53], could be at high risk of predation by the savannah cat. We assumed that the rocky outcrop habitats that were used by all *Petrogale* spp. would confer some measure of protection from predation, but it is possible that the greater size and agility of savannah cats would nullify the advantages provided by rocky habitats; all species of *Petrogale* then could be at high risk of predation by savannah cats. Similarly, although we assumed that native mammals with female body weights exceeding 18 kg would be immune to predation by savannah cats, juveniles and subadults of these species might be susceptible. For example, females of the common wombat *Vombatus ursinus* and two species of hairy-nosed wombat *Lasiorhinus* spp. achieve adult body weights that commonly exceed 20 kg, but small, young wombats would likely be vulnerable to predatory attacks by savannah cats. If these assessments were to stand, at least 15 further species of native mammals would have increased susceptibility to predation by savannah cats than the conservative totals that are shown in Table 1.

We recognise that many vertebrates other than just mammals are also likely to be more susceptible to predation by savannah cats than domestic cats, including large ground-active birds, waterfowl, frogs, large lizards, such as some species of *Varanus*, and nesting and hatchling freshwater turtles. However, while it would be possible to extend analyses to cover additional taxa, as has been done in earlier works e.g., [37,52], we have kept our focus here on mammals simply to illustrate the general approach of our decision–framework.

## 4. Summary and Conclusions

In the absence of experimental studies, such as predator removals, which show the population-level effects of predation [58], there will inevitably be debate regarding the susceptibility of native species to any predator. However, there are logistic and ethical difficulties in manipulating predator numbers, as well as rising community sentiment regarding the use of lethal removal methods [59], and these make the task of gaining strong inference about predator-impacts all the more challenging. In the case of the savannah cat, which has not established free-roaming populations, experimental studies of any kind are likely to be completely infeasible and ethically inappropriate. In the face of proposals to import this hybrid, with the attendant risk that animals will escape and might establish over large areas, a decision–framework such as that proposed here can therefore provide a simple means of predicting those components of native faunal assemblages that are most likely to be at risk.

Subjectivity in the making of assessments could be reduced by means such as expert elicitation, and confidence in the likely overall impacts following the arrival of a novel hybrid increased by adding further potential prey groups into the decision framework. For example, here we focused on non-volant native mammals in Australia, but it is likely that an established population of savannah cats would amplify the impacts that domestic cats currently have on waterfowl, large lizards, frogs, and nesting and hatchling freshwater turtles by virtue of their ability to subdue relatively large prey and their preference for hunting in densely vegetated and swampy areas. It is possible also that an established population of savannah cats would compromise the security of predator-susceptible threatened species that reside within Australia’s network of fenced conservation reserves due to their ability to jump and clear obstacles as high as 2.5 m [31]. The exclusion fences that surround these reserves are usually no more than 1.8 m high [60], but they are effective at preventing access by domestic cats and red foxes. Nineteen fenced exclosures in Australia currently cover 346 km^2^ and protect populations of 25 species of threatened native mammals that are susceptible to predation by domestic cats and red foxes [61]. The costs of raising the height of the fences that now surround these conservation havens, should savannah cats become established, would be prohibitive.

The decision in 2008 to ban importation of the savannah cat into Australia was made quickly and with very limited information on how this hybrid might affect native fauna. Our results strongly support the precautionary approach that was taken at the time, and they provide *post hoc* confirmation that a very large proportion of the extant mammal fauna of the continent would have been susceptible to predation by savannah cats if erstwhile pets escaped and established in the wild. It was a good decision, which was made with careful consideration of limited evidence and appropriate consideration of risk, by a prudent government agency, and supported by the relevant Minister. The decision–framework proposed here could be modified to identify native species that might be at risk from the importation and establishment of other hybrid cats, and potentially other novel species too. We note finally that the framework is proposed as a risk assessment tool. It could be viewed alternatively as an hypothesis, but this alternative is unpalatable: tests of predictions derived from the framework about predator-susceptibility could only be made if the savannah cat or other hybrids were to become established. The use of the decision–framework could help to preclude such a calamity from occurring.

## Figures and Tables

**Table 1 animals-09-00795-t001:** Numbers of native, non-volant Australian terrestrial mammal species (n = 246) categorised as susceptible to predation by the domestic cat ^1^ and savannah cat ^2^ (shown as numbers in parentheses as changes from values for the domestic cat), cross-tabulated with their IUCN conservation status.

Item	Level of Susceptibility to Predation					
Conservation Status (IUCN)	Extreme ^3^	High	Low	Not Susceptible	Not Assessed	Total
Extinct ^3^	19	6	1	1	3	30
Critically Endangered	2	4	3 (+1)	2 (−1)	0	11
Endangered	0	6 (+2)	4 (+1)	3 (−3)	0	13
Vulnerable	4	15 (+2)	11 (−1)	4 (−1)	0	34
Near Threatened	1	5 (+8)	18	12 (−8)	0	36
Least Concern	0	5 (+27)	90 (−12)	27 (−15)	0	122
Total	26	41 (+39)	127 (−11)	49 (−28)	3	246

^1^ Data extracted from Appendix A of Radford et al. [53], ^2^ data modified by application of the assessment criteria for savannah cats as described in the text, ^3^ neither the ‘extreme’ category of susceptibility to predation nor the IUCN category ‘Extinct’ were scored for savannah cats.

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
