# Peer review of "Assessing Risks to Wildlife from Free-Roaming Hybrid Cats: The Proposed Introduction of Pet Savannah Cats to Australia as a Case Study"

_animals, 2019, doi:10.3390/ani9100795_

Round 1

Reviewer 1 Report

This paper was a pleasure to read - well structured and well set out.  I offer a few suggestions:

Consideration should be given to including all three of the larger Dasyurus spp. as being at risk given that Sarcophilus harrisii was deemed so, and particularly as the dispersing juveniles of all three species are small at the time of independence/dispersal and incredibly predator-niaive.  Similarly, I don't understand why Petrogale lateralis was excluded given the inclusion of the other similar size rock wallaby species.  I accept that this more wide-spread taxon has populations that likely fall outside of the predicted climate range of savannah cats, but the south-western WA populations would certainly be at risk.  Given the current suite of threats facing both Vombatus ursinus and the Lasiorhinus spp. (e.g. mange) and that juveniles of all three species would be easy prey there is a case for re-assessing the two species with the more southerly distribution. The use of the term 'domestic cat' needs to reviewed in the text, as there are cases where the domestic version of the cat is being specifically referred to appropriately, and other places where the reference is actually to 'cats' in general (free roaming and domestic; e.g. Lines 213, 233, 264, 279,296, 299,302,305, 351, 357 and 359).  While this may seem a little pedantic, and most readers with a scientific background will readily exchange one term for the other, those supporters of domestic cats and certainly those advocating hybrid cats, may seize on those parts of text that appear to excuse domestic cats from the specific arguments at hand at each of those sections of the text. Consideration should be given to including reference to "nesting and hatchling freshwater turtles" after "large lizards..." in L351. Suggested edits/corrections - L50 add 'the' after '... more frequently in". L297 delete "...to domestic cat predation." L313 delete hyphen in "water-rat". L343 Suggest changing the second part of the sentence to read "..., with the attendant risk that animals will escape and may establish over large areas, ...". References - Ref 31 Latin binomial of the two felids species should be in plain text. Ref 50 Italicize "Wild Cats of the World" as it is a book title. Table S1. As discussed above consider adding the two Dasyurus spp., Petrogale lateralis and two southern wombat species.  If this suggestion is accepted the numbers and percentages in Table 1 in the body of the text, and the Abstract will require commensurate edits.

Author Response

Thank you for these comments. With respect to the inclusion of Dasyurus spp. as being at risk of predation by savannah cats, we included D. maculatus in Table S1 as it is the only member of the genus that was rated as being at greater risk of predation from savannah cats than from domestic cats. All other species were listed already by Radford et al. (2018) as being at high risk from domestic cats, and for this reason were not listed again in Table S1. Similarly, Sarcophilus harrisii was assessed as being at greater risk of predation from savannah cats than from domestic cats, and thus was entered in Table S1; its assessment went from being not susceptible to domestic cat predation to being of low susceptibility to predation from savannah cats. Petrogale lateralis, similarly, was not added to Table S1 as it had been rated already by Radford et al. (2018) as at risk from domestic cats. As we note in Table S1, "Only those mammal species assessed as more susceptible to predation by savannah cats than by domestic cats are listed here. The full list of all mammal species assessed is provided in Table S1 of Radford et al. 2018 [53]."

We appreciate that the rationale for listing species in Table S1 is not always clear, and that the species listed can be best understood by reference to Radford et al. (2018) - citation [53]. To help clarify why species have been listed in Table S1, or not, we have therefore added a comment about the Tasmanian devil at line 322, and a full paragraph at lines 328 - 344. This paragraph also discusses the assessment of wombats and the possibility that predation by savannah cats on juveniles could increase the level of their risk assessment. We hope these changes clarify both the specifics of why species were included (or not) in our assessments in Table 1 and S1.

We have also gone through the manuscript to ensure that the term 'domestic cat' is used, rather than 'cat', wherever this was possible and appropriate (e.g., lines 214, 223, 255, 281).

We have, as suggested, added reference to "nesting and hatchling freshwater turtles" at two places in the text (lines 347-8 and 374), and have also made all the minor edits/corrections suggested by the reviewer.

Reviewer 2 Report

Overall, a very interesting and worthwhile paper.

A comment regarding the title - I found it slightly confusing in that it could be interpreted that it relates to a proposal to introduce free-roaming savannah cats to Australia; an easy fix with a slight change to language.

Something that perhaps might be worthy of inclusion is a brief discussion of the fact that vulnerability of animals to predation changes at different life stages; using female body weight as an indicator of likelihood of predation may underestimate the threat to infants and juveniles in some species – this threat may vary depending on developmental and behavioural characteristics of the predated species.

Two further comments:

18-19  As per 104-105, hybrid cats may be released deliberately or accidentally from a range of sources, not only wildlife shelters.

102-103  Euthanasia may not be considered a welfare problem; I would recommend discussing both the welfare and ethical considerations here.

Author Response

Thank you for these comments. We have now amended the title to clarify that it refers to the introduction of savannah cats as pets, not as free-roaming animals.

We have also now included discussion of the possibility that juveniles of large-bodied species could be susceptible to predation by savannah cats, using wombats as an example. We briefly discuss the consequences for our results if such species were considered to be at risk of predation from savannah cats (lines 338-344).

Both of the further comments from the reviewer have been addressed (lines 18, 97, 102-104).

Reviewer 3 Report

This paper provides a thorough review of the potential impact of savannah cats to Australia's already diminishing mammal species using a well-defined decision framework and building on existing work that has been previously published. There are clearly challenges with drawing conclusions about a broad range of species where experimental studies are absent but this kind of assessment should be at the forefront of all decisions about species that pose an unknown and potentially detrimental effect on Australia's native fauna, particularly where there is a paucity of knowledge on the likely impact. As identified by the authors, it would be beneficial to provide the potential impact to other taxa, but I also acknowledge that this would be a monumental task given the possible prey species of both the savannah cat and its hybridised form. Moreover there is clear justification as to why the paper addressed only mammals. The authors also conclude that such a tool should be utilised as a risk assessment tool where rapid decision making is required by government for other hybrid cats, a well justified suggestion given the potential impacts based on their assessment.  

Author Response

Thank you for these very encouraging comments.